# The Role of Technology in the Perpetration of Childhood Sexual Abuse: The Importance of Considering Both In-Person and Online Interactions

**DOI:** 10.3390/children10081306

**Published:** 2023-07-29

**Authors:** Elizabeth L. Jeglic, Georgia M. Winters

**Affiliations:** 1Department of Psychology, John Jay College of Criminal Justice, New York, NY 10019, USA; 2School of Psychology and Counseling, Fairleigh Dickinson University, Teaneck, NJ 07666, USA; gwinters@fdu.edu

**Keywords:** child sexual abuse, online, in-person, technology, prevention

## Abstract

Child sexual abuse (CSA) is a pervasive global problem. To date, prevention efforts have largely focused on legislative efforts, parent and child education, and environmental protections. Due to the proliferation of the Internet, and especially since the COVID-19 pandemic, recent prevention efforts have focused on online CSA. However, the extent to which technology is being used in the perpetration of in-person, contact CSA remains unclear. This study examined the role of technology in the perpetration of in-person, contact CSA using a sample of 332 adult CSA survivors who completed an anonymous online survey. Overall, we found that only 8.5% of the sample reported that they met the perpetrator online through social media, chatrooms, and other online applications. When looking at the role of technology in the perpetration of the abuse, 35% reported texting with the perpetrator, 27% reported engaging in online chats, and 33% spoke to the perpetrator on the phone. Few participants reported sending (11%) or receiving (13%) photographs or videos that were sexual in nature. Technology use was reported more frequently in CSA involving adolescents than CSA involving children aged 12 and under. There were no differences in the use of technology based on the age of the person who perpetrated the CSA. These findings will be discussed as they pertain to prevention efforts for CSA.

## 1. Introduction

Childhood sexual abuse (CSA) is a pervasive global problem. Overall, it is estimated that 127 out of 1000 children (7.8% boys; 18% girls) will experience contact sexual abuse by the time they reach age 18 [1,2], although prevalence rates of CSA vary depending upon the reporting source (e.g., self-report versus official report), the sex of the child, and the global region. The scope of CSA is extensive, with numerous negative long-term consequences to the victim, their family, and the community [3]. Yet, it has been labeled a global health crisis that is preventable [4], and as such, significant resources have been dedicated to preventing CSA. 

In recent years, and especially since the COVID-19 pandemic, there has been increased government and public attention on the prevention of online sexual abuse (e.g., online solicitation, child-sexual-abuse material (CSAM)). For example, in April 2020 the National Center for Missing and Exploited Children (NCMEC) received four times as many cybertips for online CSA as in April of 2019 (4.1 million versus 1 million; [5]). This has resulted in increased legislation focused specifically on online sexual abuse (e.g., [6]). However, the scope of behaviors that constitute online sexual abuse is broad, ranging from inappropriate sexual talk or comments, sharing and disseminating sexual images, use of online platforms to meet a child in-person, and the creation and dissemination of CSAM [7,8,9]. Further, there is evidence that a significant amount online CSA may be perpetrated by an individual who was known to the child offline [10]. That is, the traditional stereotype of a stranger lurking online seeking to abuse a minor child may not be reflective of what actually transpires online [11]. Consequently, there has been little attention paid to how online and electronic communication may be used to facilitate CSA, or in the course of in-person, contact CSA. Thus, the goal of this study was to investigate the role of technology in the perpetration of in-person contact CSA based on self-report data from 332 survivors who experienced CSA before the age of 18, including exploring differences based on the age of the child victim and the person perpetrating the CSA. 

## 2. Online Sexual Solicitation and Sexual Grooming

The online sexual solicitation of minors has been steadily increasing as more children have access to the Internet and Internet-enabled devices [12]. While there is no universal definition of online sexual solicitation, it has generally been used in the literature to refer to unwanted or nonconsensual sexual talk, questions, or requests for sexual acts by an adult and/or a youth (e.g., [10]). This has caused some confusion, as terms such as online solicitation, online sexual abuse, and online sexual grooming have often been used interchangeably. Finkelhor and colleagues [10] defined online solicitation as:

“Unwanted solicitations (unwanted sexual talk, unwanted sexual questions, and unwanted sexual act requests) by adults or other youth (known or unknown)” (p. 4, 11). 

However, they indicated that “because so many of these episodes were brief and of unknown source, only those from known adults are included in subsequent categories” in which they further broke down the category of online solicitation to include online child sexual abuse and online grooming by an adult [10]. Indeed, online sexual solicitation only needs to occur once with minimal interaction with the minor, whereas online sexual grooming is a process that involves the development of an intimate relationship with the child [13]. While a universal definition of online sexual grooming has yet to be proffered, it refers broadly to the process involving victim selection, developing trust and building rapport with the victim, and desensitizing the minor to sexual content prior to initiating the online CSA [14,15]. Thus, not all sexual solicitation involves sexual grooming and can simply refer to a perpetrator seeking sexual gratification without establishing any bond or relationship with the minor [13]. 

### Prevalence

Existing research shows varying rates of online sexual solicitation ranging between 17 and 32% of those under the age of 18 (e.g., [8,13,16,17]). A recent meta-analysis found that approximately 20% of individuals under the age of 18 have experienced unwanted sexual exposure online (defined as “exposure to sexually explicit pictures and/or videos via pop-up windows, spam e-mails, Web site links, etc., without seeking or expecting sexual material” p. 13, [18]), with 11% reporting sexual solicitation (defined as “requests to engage in unwanted sexual activities or sexual talk, or to provide sexual information to another individual (including peers and adults)” [18], p. 134).

Other studies have looked at prevalence rates for online sexual grooming, with rates ranging from 9.6% to 29.8% depending upon the age and gender of the child [19,20,21]. A study of 1,704 minors (12–15 years old) in Spain found that between 7% and 18% endorsed various online grooming strategies as measured by the Multidimensional Online Grooming Questionnaire [22]. 

Finkelhor and colleagues [10] conducted an online survey of a representative sample of 2639 U.S young adults (aged 18–29) about the online and technology-facilitated sexual abuse they experienced before the age of 18. Overall, they concluded that a substantial portion of their sample reported some form of online CSA. While there was considerable overlap in the definitions of the abusive behaviors used to assess prevalence rates, 22.5% of respondents reported online solicitation (i.e., unwanted sexual talk, questions, or sexual act solicitation by other youth and adults), 11% reported image-based sexual abuse (i.e., nonconsensual taking of images, sharing or forcing/threatening to share images in a statutorily impermissible relationship), 15.6% reported online CSA (i.e., unwanted solicitation and image-based sexual abuse from adults only), 7.2% reported nonconsensual sexting, and 5.4% reported online grooming (see [10] for a complete list and description of online offenses and definitions). 

## 3. The Role of Technology in Offline Sexual Abuse

There is limited research examining how much online sexual solicitation results in offline CSA. In one of the few studies that have examined the progression from online to offline contact, Finkelhor and colleagues [8] found that 19% of their sample of 1501 youth between the ages of 10 and 17 reported online solicitation, but only 3% involved attempted or had actual offline contact by telephone, by mail, or in-person. In a scoping review, Chauviré-Geib and Fegert [23] used the term technology-assisted child sexual abuse (TA-CSA) as “a collective term to describe the use of technology in abuse” (p. 2), and they only found three qualitative studies that described the progression from online to offline sexual grooming and abuse in some of the cases [24]. While sex trafficking of minors is often seen as distinct from CSA, there is increasing evidence that minors are being recruited online for offline trafficking. Recent research found that of those minors that were trafficked in 2015 or later, 55% were recruited via text, website, or app [25]. 

In their study assessing the prevalence of online sexual offenses against children in the U.S., Finkelhor and colleagues [10] found that a significant number of participants who reported online sexual abuse also knew the perpetrator offline—ranging from 23.5% of those who experienced online solicitation to 79.5% who reported online sexual grooming. The findings of this study contrast with the perception that those who perpetrate online sexual abuse are strangers and predators who manipulate and threaten young children (see [11,26]). In line with this perspective, Greene-Colozzi and colleagues [13] found that 23% of their sample of 1113 college undergraduates engaged in long intimate conversations with adult strangers (consistent with sexual grooming). Further, 38% of those individuals met the adult stranger in-person when they were minors, with 68% reporting that they engaged in physical sexual intercourse with the adult. 

In addition to meeting and accessing their victims online, technology can also be used as part of the sexual-grooming process for offline sexual abuse. In-person child sexual grooming has been defined as “the deceptive process used by sexual abusers to facilitate sexual contact with a minor while simultaneously avoiding detection.” [27]. Winters and colleagues [28] identified 42 distinct sexual-grooming behaviors that have been identified by experts to constitute behaviors indicative of sexual grooming (see [28] for review). Several of these sexual-grooming behaviors and tactics can be engaged in using technology such as social media, e-mail, texting, and videochatting. For example, technology can be used to show the child pornography images or videos, whereas texts, chats, or conversations can involve inappropriate sexual language or jokes. Technology can also be used to communicate frequently and affectionately with the minor, or to psychologically isolate the minor from their peers and guardians. 

One of the only studies that has examined how online sexual grooming relates to offline CSA was conducted by Shannon in 2008. He reported on police data from Sweden in which police-reported CSA involved online contact between the perpetrator and the victim. Of the 315 cases, 57% involved only online contact; in 22% of the cases the perpetrator met the minor online but the abuse happened offline; in 14% of the cases the perpetrator and victim were in contact both online and offline but it was not clear if any abuse took place offline; in 7% of cases the perpetrator already knew the victim offline but used the Internet to sexually groom the victim. 

## 4. Youth, Technology, and Sexual Abuse

There has been increasing evidence that a significant proportion of sexual offenses against children are perpetrated by another minor. It is estimated that those under the age of 18 account for between 30 and 50% of all cases of CSA [29,30]. Given that 95% of young people have access to Internet-enabled devices [31] and teens prefer online communication and texting with their peers and romantic partners [32,33], it is likely that at least some youth problematic sexual behavior takes place online. In their 2022 study of the prevalence of online sexual abuse against minors in the U.S., Finkelhor and colleagues reported that in cases where the age of the perpetrator was known, youth (those younger than 18) comprised between 35 and 52% of the perpetrators. 

As the use of technology for communication such as texting and social media increases as children enter adolescence [34], it is also likely that technology may play a more significant role in CSA. For example, in their study Finkelhor and colleagues [10] found that across all online sexual-abuse categories, 16% or less of the sample reported being 12 years of age or under at the time of the online sexual abuse. Further, Winters and colleagues [28] speculate that sexual grooming behaviors may differ based upon the age of the child, so it is likely that technology may play a larger role in the sexual abuse of adolescents as opposed to those who are age 12 and under. 

## 5. Current Study

With increased recognition of the role of the Internet in CSA, many countries have enacted legislation to make the luring of children via electronic means a crime [35]. However, there is inconsistency in the ways that researchers have defined online sexual solicitation and there is evidence that prevention strategies focusing primarily on the online environment may be misdirected, as many youth who experience online sexual solicitation and/or online sexual grooming may actually know their perpetrator in an offline environment [10,11]. Further, technology may more broadly play a role in CSA as part of the sexual-grooming process, but there has been little empirical investigation to date. Thus, the purpose of the current study was to examine the role of technology in the perpetration of CSA among a sample of adults who reported a history of unwanted sexual contact before the age of 18. This included examining both online and offline forms of electronic communication, including text, email, online chats, and phone and video conversations. We also examined the sending and receiving of photographs or videos, and whether those were sexual in nature. Based upon previous research, it was anticipated that most of the individuals who reported a history of CSA would not meet their perpetrators online, but that technology use may have played a role in the CSA, especially among adolescents.

## 6. Method

### 6.1. Participants and Procedure

Participants were 332 adult volunteers who were a subsample of a larger study of individuals (*n* = 674) who reported that they had experienced unwanted sexual contact (sexual abuse) before the age of 18. To participate in the larger study, individuals had to be age 18 years or older, speak and write in English, and currently reside in the United States. For the present analyses, only participants who were sexually abused in 2001 or after (i.e., when the Internet was present in 50% of private homes [36]) were included, as estimated by the age at time of participation and age at time of the sexual abuse. The online-survey participants completed a range of questions related to their experience of sexual abuse as a minor. This was done through the research platform, Prolific. Participants were paid USD 5 for the completion of the confidential 40 min survey. Selected questions from this survey were included in the present analyses, including items related to demographics (e.g., age, sex, race/ethnicity), characteristics of the abuse, and contact with the person who committed the abuse through online and electronic communications. Upon completion, the participants reviewed a debriefing form regarding the purpose of the research, contact information for the researchers, and contact information for emergency mental health resources. 

Participant responses (*n* = 68) from the larger study were removed if they were incomplete, the participant did not meet inclusion criteria (e.g., they indicated their age at the time of the CSA was 18 or above), or the participant failed more than one of four manipulation checks; this resulted in a database of 674 participants. For 332 of these participants, CSA had occurred in the year 2001 or after (49.26% of the original sample). Participants’ ages ranged from 18 to 37 years (*M* = 25.81 years). There were 259 (78.0%) participants who identified as biological females and 68 (20.5%) as biological males, 3 (0.9%) defined their sex at birth as “other”, and 2 (0.6%) preferred not to answer. Most participants reported their self-identified gender as female (*n* = 299; 69.0%), with 19.6% (*n* = 65) as male, 7.2% (*n* = 24) as non-binary, 3.0% (*n* = 10) as transgender, and 1.2% (*n* = 4) as “other”. The sample identified as primarily White (*n =* 208; 62.7%), with 11.1% (*n =* 37) identifying as Black, 8.7% (*n =* 29) as Hispanic or Latinx, 8.1% (*n* = 27) as Asian/Pacific Islander, 8.7% (*n* = 29) as biracial or multiracial, and 0.3% (*n* = 1) as “other”; 1 preferred not to answer (0.3%). Half of the sample identified as heterosexual (*n* = 167; 50.3%), while there were 91 (27.4%) who identified as bisexual, 30 (9.0%) as pansexual, 27 (8.1%) as homosexual, 9 (2.7%) as asexual, and 6 (1.8%) as “other” sexuality; 2 (0.6%) preferred not to answer. There were 135 (40.7%) individuals who were single or in a casual relationship, 128 (38.5%) who were in a committed relationship or domestic partnership, and 67 (20.2%) were married; 2 (0.6%) preferred not to answer. 

### 6.2. Variables

#### 6.2.1. Characteristics of the Sexual Abuse

Participants responded to a question about how many individuals committed sexual abuse against them (“How many different people have you had unwanted sexual contact with before you were 18 years old?”); for those who experienced more than one victimization, they responded to the remainder of the questions about the individual they had the most interpersonal contact with. Participants reported information about themselves (i.e., “How old were you when the unwanted sexual contact started?”; “Please indicate your relationship with the individual who you had the unwanted sexual experience with.”) and the person who committed the abuse (i.e., “How old was the offender when the unwanted sexual contact started?”; “What was the offenders’ gender?”) at the time of the abuse. 

#### 6.2.2. Electronic Communication

Participants reported whether they met the individual who committed the sexual contact online or in-person (i.e., “How did you meet the offender?” with response options “Online,” “In-person,” or “I would rather not say”). Further, they indicated (yes/no/prefer not to answer) to whether they had text conversations, email exchanges, online chats, or phone conversations with the person (i.e., “Did you have text conversations/email exchanges/online chats/phone conversations with the person who you had unwanted sexual contact with?”). They were also asked whether they sent or received photographs and videos with the individual (i.e., “Did you send them photographs or videos of yourself?”; “Did you receive photographs or videos of the person?”), and whether those were sexual in nature (i.e., “Were any of the photographs or videos sexual in nature?”). 

### 6.3. Analytic Plan

The analyses included descriptive statistics (i.e., frequency and percentage) for all variables. Chi-square analyses were run to explore any differences in the electronic communication based on the age of the victim (children aged 12 or below; adolescents aged 13 or above) or the individual who committed the offense (minors aged 17 or below; adults aged 18 or above). In instances where any cell count was less than five, a Fisher’s exact test was conducted.

## 7. Results

Approximately half of participants (*n* = 188; 56.6%) experienced CSA by one individual, while the remaining 41.9% (*n* = 139) reported abuse by more than one individual; 5 (1.5%) preferred not to answer. As noted above, for the individuals who experienced CSA by more than one individual, they responded to the remaining questions about the person they had the most interpersonal contact with. The average age of the victim at the time the abuse began was 13.01 years (range = 3–17), while the average age of the person who perpetrated the abuse was 23.39 (range = 4 to 65). Around two-thirds of the sample (*n* = 216; 65.1%) were aged 13 to 17 at the time of the abuse, while the others (*n* = 116; 34.9%) were aged 12 or below. Results showed 59.9% (*n* = 199) of the individuals who committed the abuse were adults, while 37.0% (*n* = 123) were minors themselves (age 17 or below); 10 preferred not to answer (3.0%). Regarding the relationship between the participant and the person who committed the offense, the person was most often a romantic partner (*n* = 62; 18.7%), friend (*n* = 62; 18.7%), friend of the family (*n* = 53; 16.0%), or extended family member (*n* = 39; 11.7%).

### 7.1. Electronic Communication 

As seen in Table 1, most participants indicated they met the person who committed the offense in-person (*n* = 303; 91.5%), with only 8.5% (*n* = 28) people reporting they first met the person online. Chi-square results showed participants who were aged 13 to 17 at the time of the abuse were significantly more likely to report they met the person who committed the abuse online than those who were abused at age 12 or below. The was no difference when comparing the age of the person who perpetrated the abuse (minors aged 17 or below; adults aged 18 or above).

For the 324 cases who reported responses for each electronic-contact item, results showed that approximately half of participants (*n* = 156; 48.1%) reported they did not have any online or electronic communications (i.e., text, online chat, phone, or video conversations) with the person who committed the CSA. There were 168 (51.9%) who reported engaging in one or more types of online or electronic communication; 54 (16.7%) reported one type, 47 (14.2%) reported two types, 40 (12.3%) reported three types, 23 (7.1%) reported four types, and 4 (1.2%) reported all five types. 

Table 2 contains information regarding the characteristics of the CSA reported by participants. Approximately one-third of participants reported they engaged in text (*n* = 115; 34.8%) or phone (*n* = 109; 33.2%) conversations with the individual. Further, 27.3% (*n* = 90) of the sample reported engaging in online chats. Few participants reported engaging in video conversations (*n* = 30; 9.1%) or email exchanges (*n* = 24; 3.6%) with the person who committed the CSA. There were 63 participants (19.3%) who sent photographs or videos to the person who perpetrated the sexual abuse. Of the 48 individuals who responded to the question, 35 (72.9%) reported that these photos or videos were sexual in nature. Further, 23.7% (*n* = 76) of participants received photographs or videos from the person who perpetrated the sexual abuse. There were 57 participants who responded to whether these were sexual in nature, with 46 (80.7%) confirming that they contained sexual content.

### 7.2. Differences Based on Age of the Victim and Person Who Perpetrated the Abuse

As seen in Table 2, results showed that adolescent victims were significantly more likely to have engaged in most forms of electronic communication compared to child victims, including text, online chat, video, and phone conversations. The only comparison that was not significant was for email exchanges, with both groups showing low frequency of endorsement overall (3.4% for children and 7.5% for adolescents). Further, adolescent victims were also more likely to report they sent or received photographs or videos with the person who committed the offense compared to child victims, but there was no difference between groups regarding whether they were sexual in nature. Regarding the age of the person who committed the offense, there were no differences in the use of electronic communication when the person who committed the abuse was a minor or an adult. 

## 8. Discussion

Since the COVID-19 pandemic, there has been considerable media and legislative attention paid to the problem of online sexual abuse. While there is evidence that the number of individuals who are seeking to create CSAM is increasing [37,38] and online sexual solicitation and grooming is a serious issue [39], recent research suggests that a significant number of children who experience online sexual abuse know the person offline [10]. As such, this study examined the role of technology in the perpetration of CSA among a sample of adults who reported unwanted sexual contact before the age of 18 that occurred in 2001 or after (i.e., when the Internet was present in 50% of households). Overall, we found that very few adults (8.5%) who reported a history of CSA met their perpetrator online. One-half of participants reported engaging in at least one type of electronic communication with the person who committed the abuse, with texting (34.8%), phone (33.2%), and online chats (27.3%) being most common. Further, the use of technology both to meet, and then to perpetrate the contact abuse, was generally reported more frequently by adolescents as opposed to children aged 12 and under. There were no differences in the use of electronic communication based on whether the person who perpetrated the abuse was a minor or an adult. However, given that we did not have information as to whether the individuals in this sample had Internet or cell-phone access at the time they experienced the CSA, the results of this study should be interpreted with caution as described below in the limitations.

As hypothesized, we found that less than 10% of our sample reported that they met their perpetrator online. This is lower than rates reported by Shannon [16], who found that in 22% of Swedish police reports of online sexual abuse, the perpetrator met the victim online but perpetrated the sexual offense offline. However, it should be noted that Shannon’s study was based upon official police reports while this study used self-report to assess CSA. Given the notoriously low rates of reporting of CSA to authorities (e.g., [40]) it is likely that Shannon’s findings reflect a different population than that of this study as the majority of those in our sample did not report their CSA to law enforcement. Further, it is unclear how many participants in our sample, and particularly those under the age of 12 at the time of the CSA, had Internet access at the time of the abuse.

While there is evidence that child sex traffickers are increasingly using online means to lure minors into sex trafficking [25], little research has examined the role of the Internet in offline CSA. Finkelhor and colleagues [10] found that almost 80% of minors who experienced online sexual grooming also knew their perpetrator offline, but they did not assess whether CSA was perpetrated offline. Ultimately, our results showed that while online CSA is undoubtedly important to address, the vast majority of contact CSA is being committed by individuals who met the child in-person, not in an online setting. 

We also found that various forms of technology played a role in the perpetration of the abuse. One-half of participants reported engaging in at least one type of electronic communication with the person who committed the abuse, although some types were more common than others. One-third of the sample reported speaking to the perpetrator through phone (33.2%) or text (34.8%) messages, and one-quarter (27.3%) engaged in online chats with the person. While we did not assess the details of the communications (e.g., content, frequency), it is likely these were being used to facilitate the abuse process (i.e., sexual grooming). There is evidence that technology, such as texting, has been used by those employed by youth-serving organizations to perpetrate sexual abuse, as it gives the individual who wishes to perpetrate the abuse constant access to the victim and enables them to circumvent guardianship as they can access minors directly through their phones [41]. 

As expected, results showed that adolescents (those aged 13 and over at the time of the abuse) were more likely to report meeting the individual who perpetrated the CSA online as opposed to children aged 12 and under. While few participants in each group (under 12 and over 13) reported meeting their perpetrator online, it is logical that online methods would be used more frequently with adolescents given their increasing usage of technology for communication and socialization [34]. Wolak and colleagues [11] also suggest that contrary to the “online predator” stereotype that has been perpetuated by the media, adults that use the Internet to meet minors online use the guise of romance and friendship to meet and abuse minors, particularly adolescents. That was also reported by Greene-Colozzi and colleagues [13] who found that two-thirds of individuals who reported meeting adult strangers online and then having intercourse with them offline considered the sexual intercourse to be consensual at the time. While we did not have information about the context and circumstances under which the participants in this study met the perpetrator online, it would be important to have a more nuanced understanding of the tactics and behaviors used first to entice the minor online and then the process by which the offline CSA occurred. Importantly, we also found nearly all of the electronic communication types (i.e., text, phone, video, and online conversations) were happening significantly more frequently with teenagers (those aged 13 and over) as compared to children aged 12 and under. This can be more worrisome for parents as it is easier to monitor younger children’s online presence and behavior. However, as children enter adolescence parents are less likely to monitor their online behavior [42]. It may be beneficial for future research to get youth input into strategies that adults and guardians can use to keep them safe from CSA when using technology.

Interestingly, while more than one-third of the individuals who perpetrated the CSA in this sample were under the age 18 (37%), the use of technology in the perpetration of the abuse did not significantly differ when the person who committed the abuse was an adult or adolescent. This is somewhat curious as young people tend to communicate with one another using text or social-media apps [34], so it would follow that they may also tend to use technology when they engage in problematic sexual behavior. However, in our sample, 44% of adolescents who engaged in problematic sexual behavior had victims who were under the age of 12. Younger children may have increased parental oversight of their online communications and generally have less access to cellphones than adolescents. Further, youth who engage in problematic sexual behavior may abuse other children in the home such as siblings, and thus they will not need electronic means to access them or communicate with them [43].

### 8.1. Limitations

This study is not without limitations. First, as this was an online self-report study, we cannot verify the accuracy of the reports. However, there is evidence that anonymous self-report may yield more accurate responding as it is a sensitive topic that is believed to be significantly under-reported to authorities [1,44]. It should also be noted that we focused on age-related variables for the present analyses, but there may be other demographic factors or abuse characteristics that would influence the extent to which technology is used in the CSA process. For example, our sample was primarily female (78%) and White (63%); it may be that these variables, among others (e.g., socioeconomic status), could also impact the use of technology and therefore, could be explored in future studies. Moreover, we included CSA instances from 2001 and beyond (i.e., when there were 50% of households that had the Internet) in the data analyses, however, we did not have information about whether participants in this study had access to the Internet at the time of the CSA and thus this limitation should be underscored when interpreting the findings of this study. Furthermore, as Internet and cell phone usage by young people, especially since the COVID-19 pandemic, has steadily increased and the types and functions of online applications and safeguarding protections are constantly changing, these data do not necessarily reflect CSA as it is being perpetrated at the present time. While it is likely that the findings observed in the current study will be relevant today, additional research using adolescent samples is needed to make this conclusion. 

### 8.2. Implications

There are several implications for the findings of this study. First and foremost, there is a pressing need to further explore the use of technology in the perpetration of offline CSA. As youth increasingly use technology to socialize and communicate, it is inevitable that it will also play an increasingly important role in the perpetration of CSA. Understanding how perpetrators are using technology is pivotal for prevention of CSA. As we are now starting to understand which sexual-grooming behaviors may be red flags for offline CSA [45], it is important to understand how those behaviors are manifested technologically for prevention efforts. This can enable guardians to recognize potential technologically-facilitated sexual-grooming behaviors before the CSA takes place. For instance, companies have already developed software that can scan children’s emails, texts, social media, and apps for harmful interactions and content (e.g., bark.us; thorn.org). While these programs can detect sexually explicit images and language, it is unclear if they may pick up some of the more subtle sexual-grooming behaviors that could be manifested online, especially behaviors that take place in the trust-development phase, such as praise, compliments, attention, and affection. Our study showed that these types of prevention efforts are especially important for adolescents, as they are more likely to engage in electronic communication with the person who committed the abuse compared to children. While we know that certain youth may be at increased risk of online sexual abuse (i.e., those who identify as sexual or racial/ethnic minorities and those who are disabled) [46], more research on risk factors and how to target online prevention efforts for specific at-risk groups is needed. 

While there has understandably been increased legislation focused specifically on online sexual abuse (e.g., [6]), the results of this study showed that many of the CSA-related offense processes may be occurring offline and most individuals in this study reported no electronic communication with the person who perpetrated the abuse. Many legislative efforts related to sexual grooming have focused on the online and electronic methods [47], but this study further highlights that laws should encompass both electronic and in-person means of facilitating the sexual-abuse process [48].

Most online child sexual-abuse prevention has been focused on adults as perpetrators; however, the findings of this study bolster some previous research suggesting that youth are also using the Internet and technology to engage in problematic sexual behaviors. Given that youth primarily communicate with one another online it is important to also address how young people may also be using the Internet to engage in sexual harm. This may be particularly salient as, given the frequency with which young people communicate using technology, it may not immediately seem worrisome if a youth sends another youth sexually explicit material online. In fact, Kaylor and colleagues [49] concluded that when young people send solicited or unsolicited nude or semi-nude pictures of themselves, it is in most cases part of modern-day flirting or courtship behavior. Therefore, it is important to educate young people, guardians, and those who work with youth about how technology can be used to perpetrate CSA, and what technologically facilitated sexual-grooming behaviors may look like when the person who engages in the problematic behavior is a youth. 

In sum, few individuals who experienced CSA met the person who committed the offense online. However, different forms of electronic communication may nonetheless be used to facilitate the abuse process in cases of in-person, contact sexual abuse. Indeed, the results highlight the importance of ongoing support for prevention, research, and legislation designed for both online and in-person sexual offenses. 

## Figures and Tables

**Table 1 children-10-01306-t001:** How the victim met the person who committed the offense.

Item	Age of Victim				Age of Person Who Committed the Offense				Total
	≤12	≥13	*x* ^2^	*p*	≤17	≥18	*x* ^2^	*p*	
**Met Person**									
**Valid N**	115	216	3.85	0.05 *	123	198	1.92	0.166	**331**
**Online**	5 (4.3)	23 (10.6)			7 (5.7)	20 (10.1)			**28 (8.5)**
**In-Person**	110 (95.7)	193 (89.4)			116 (94.3)	178 (89.9)			**303 (91.5)**

* = *p* ≤ 0.05.

**Table 2 children-10-01306-t002:** Use of technology in CSA.

Item	Age of Victim				Age of Person Who Committed the Offense Age				Total
	≤12	≥13	*x* ^2^	*p*	≤17	≤18	*x* ^2^	*p*	
**Text Conversations**									
**Valid N**	115	215	45.28	<0.001 *	123	197	1.41	0.234	**330**
**Yes**	20 (17.4)	120 (55.8)			57 (46.3)	78 (39.6)			**115 (34.8)**
**No**	95 (82.6)	95 (44.2)			66 (53.7)	119 (60.4)			**215 (65.2)**
**Email Exchanges**									
**Valid N**	116	214	n/a ^a^	0.226	122	198	0.136	0.713	**672**
**Yes**	4 (3.4)	16 (7.5)			8 (6.6)	11 (5.6)			**24 (3.6)**
**No**	112 (96.6)	198 (92.5)			114 (93.4)	187 (94.4)			**638 (96.4)**
**Online Chats**									
**Valid N**	116	214	55.15	<0.001 *	123	197	0.002	0.964	**330**
**Yes**	15 (12.9)	75 (35.0)			34 (27.6)	54 (27.4)			**90 (27.3)**
**No**	101 (87.1)	139 (65.0)			89 (72.4)	143 (72.6)			**240 (72.7)**
**Phone Conversations**									
**Valid N**	116	212	18.51	<0.001 *	116	199	0.319	0.572	**328**
**Yes**	21 (18.1)	88 (41.5)			37 (31.1)	68 (34.2)			**109 (33.2)**
**No**	95 (81.9)	124 (58.5)			82 (68.9)	131 (65.8)			**219 (66.8)**
**Video Conversations**									
**Valid N**	115	215	n/a ^a^	0.002 *	121	199	1.11	0.291	**330**
**Yes**	3 (2.6)	27 (12.6)			8 (6.6)	20 (10.1)			**30 (9.1)**
**No**	112 (97.4)	188 (87.4)			113 (93.4)	179 (89.9)			**300 (90.9)**
**Send Photographs or Videos to Offender**									
**Valid N**	114	213	22.06	<0.001 *	122	195	0.008	0.931	**327**
**Yes**	6 (5.3)	57 (26.8)			23 (18.9)	36 (18.5)			**63 (19.3)**
**No**	108 (94.7)	156 (73.2)			99 (81.1)	159 (81.5)			**264 (80.7)**
**Sexual in Nature**									
**Valid N**	4	44	n/a ^a^	1.00	16	29	2.70	0.102	**48**
**Yes**	3 (75.0)	32 (72.7)			9 (56.3)	23 (79.3)			**35 (72.9)**
**No**	1 (25.0)	12 (27.3)			7 (43.8)	6 (20.7)			**13 (27.1)**
**Receive Photographs or Videos to Offender**									
**Valid N**	113	208	29.49	<0.001 *	118	194	0.99	0.320	**321**
**Yes**	7 (6.2)	69 (33.2)			24 (20.3)	49 (25.3)			**76 (23.7)**
**No**	106 (93.8)	139 (66.8)			94 (79.7)	145 (74.7)			**245 (76.3)**
**Sexual in Nature**									
**Valid N**	4	52	n/a ^a^	0.575	16	38	n/a ^a^	0.459	**57**
**Yes**	4 (100.0)	41 (78.8)			12 (75.0)	32 (84.2)			**46 (80.7)**
**No**	0 (0.0)	11 (21.2)			4 (25.0)	6 (15.8)			**11 (19.3)**

^a^ = Fisher’s exact test used due to cell count(s) being below five. * = *p* ≤ 0.05.

## Data Availability

Data are available upon request to the corresponding author.

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
