# Peer review of "The Role of Technology in the Perpetration of Childhood Sexual Abuse: The Importance of Considering Both In-Person and Online Interactions"

_children, 2023, doi:10.3390/children10081306_

Round 1

Reviewer 1 Report

Thanks so much for the opportunity to review this paper. I have two main concerns:

1. The authors stated in the limitations section that the mean age of the participants was 35, this means that many of them may not have had access to the internet before 12 years old, and therefore precludes them from the study. I feel that this is a serious methodological flaw. By right, the authors should have asked the participants, as a part of the exclusion criteria, whether they had internet access before age 12. If they did not have access to the internet before age 12, then they should be excluded from this study.

To overcome this limitation, may I suggest that the authors stratify the respondents by age group, and examine whether there is a significant association between proportion of age group and meeting the perpetrator online. I am curious whether most of the 4.5% of the participants who met the perpetrator online belonged to those below a certain age group? I am quite certain that the participant who is 77 years old would not have met the perpetrator through online means at 12 years old or below, because the online means of communication simply did not exist. Thus the logic of this study is flawed.

2. I feel that it is good for the authors to provide a table for demographics. In Table 1, much of the information pertains CSA characteristics, which is not related to the research objectives. While interesting, I think they might be more suitable to be presented in another paper with a relevant research objective.

The data presented by Shannon 2008 is an interesting point for comparison. The high prevalence of meeting the perpetrator online only (57%) and the 22% of those meeting online but abused offline is higher than the prevalence found in this study deserves more attention in the discussion section.

Minor issues. Perhaps a few sentences could be refined in its expression.

Reviewer 2 Report

The aim of the paper is to study the role of technology in the perpetration of in contact child sexual abuse. The study is based on a sample of adult survivors who completed an anonymous online survey.  The paper explores a topic that raises serious concerns due to the identified increasing trends of online sexual abuse of children and adolescents in the recent period.   My suggestions to the authors are the following:

1. The survey questions that register sexual abuse should be described in more detail, e.g. how the measure ‘Number of Abusers’ is constructed, ‘Abusive Incidents’, etc. Overall, Section 5.2. should include more information about the questions and the response options recording characteristics of sexual abuse and electronic media communication.

2. There should be a section describing the methods used for the data analysis – descriptive stats, Fisher’s exact test, etc.

3. The authors focus the analysis on the differences based on age of the victim and person who perpetrated the abuse. It is not clear if other dimensions were explored and what the results show (e.g. gender and racial differences or other sociodemographic characteristics). The descriptives show that most of the participants are/identify as female. They authors should add a comment how this may affect the results.

4. The age of the participant ranges from 18 to 77 years. Online abuse is a relatively recent phenomenon and it is not clear how it relates to the cases of respondents at very high age. The role of technology in the perpetration of childhood sexual abuse would be stronger in the recent period (younger age groups) when the access and use of online resources and social media substantially increased. The authors should a note about age differences, if any, in the respondents’ reports of sexual abuse.

5. The authors should clearly define in the aim of the paper. i.e. what type of technology they would explore (online resources only).

6. In the Discussion part of the paper it would be informative for the readers if certain risk groups of children/adolescents are outlined. Based on this, the section on the implications should focus on the identified risk groups and the interventions they require.

Round 2

Reviewer 1 Report

Thank you again for the opportunity to submit a review. First of all, I'd like to thank the authors for the major changes they have done to the manuscript, and for adding to the limitations of the study. However, it still needs to be underscored that the interpretation of the results should take into consideration that data on the participants' access to the Internet was not recorded. This should not only be presented in the limitations section, but also in the discussion section where the prevalence is noted. Therefore, please add this fact after the following sentence, that due to the absence of this information, the results should be interpreted with great caution: "Given the notoriously low rates of reporting of CSA to authorities [e.g.,48] it is likely that Shannon’s findings reflect a different population than that of this study." - Please elaborate more of what you mean by "this population" here, and mention it could not be ascertained how many of them had internet access before age 12.

Reviewer 2 Report

The authors addressed the proposed changes in the previous review. The responses and clear and convincing and the revisions improved the analysis. The revisions in the empirical analysis make it more coherent (i.e. limitation of the data to cases of CSA that occurred in 2001 or later, as 50% of households in the United States had Internet in the home at this time).  This is why, the paper should be considered for publication.